# Leveraging Data Quality to Better Prepare for Process Mining: An Approach Illustrated Through Analysing Road Trauma Pre-Hospital Retrieval and Transport Processes in Queensland

**DOI:** 10.3390/ijerph16071138

**Published:** 2019-03-29

**Authors:** Robert Andrews, Moe T. Wynn, Kirsten Vallmuur, Arthur H. M. ter Hofstede, Emma Bosley, Mark Elcock, Stephen Rashford

**Affiliations:** 1School of Information Systems, Queensland University of Technology (QUT), Brisbane 4000, Australia; m.wynn@qut.edu.au (M.T.W.); a.terhofstede@qut.edu.au (A.H.M.t.H.); 2Institute of Health and Biomedical Innovation and School of Public Health and Social Work, Queensland University of Technology (QUT), Brisbane 4059, Australia; k.vallmuur@qut.edu.au; 3Jamieson Trauma Institute, Royal Brisbane and Women’s Hospital, Metro North Hospital and Health Service, Brisbane 4029, Australia; 4Queensland Ambulance Service (QAS), Brisbane 4034, Australia; emma.bosley@ambulance.qld.gov.au (E.B.); stephen.rashford@ambulance.qld.gov.au (S.R.); 5Retrieval Services Queensland (RSQ), Brisbane 4000, Australia; mark.elcock@health.qld.gov.au

**Keywords:** process mining in healthcare, methodologies and best practice for PODS4H, data quality, pre-hospital transport and care, GEMS, HEMS

## Abstract

While noting the importance of data quality, existing process mining methodologies (i) do not provide details on how to assess the quality of event data (ii) do not consider how the identification of data quality issues can be exploited in the planning, data extraction and log building phases of any process mining analysis, (iii) do not highlight potential impacts of poor quality data on different types of process analyses. As our key contribution, we develop a process-centric, data quality-driven approach to preparing for a process mining analysis which can be applied to any existing process mining methodology. Our approach, adapted from elements of the well known CRISP-DM data mining methodology, includes conceptual data modeling, quality assessment at both attribute and event level, and trial discovery and conformance to develop understanding of system processes and data properties to inform data extraction. We illustrate our approach in a case study involving the Queensland Ambulance Service (QAS) and Retrieval Services Queensland (RSQ). We describe the detailed preparation for a process mining analysis of retrieval and transport processes (ground and aero-medical) for road-trauma patients in Queensland. Sample datasets obtained from QAS and RSQ are utilised to show how quality metrics, data models and exploratory process mining analyses can be used to (i) identify data quality issues, (ii) anticipate and explain certain observable features in process mining analyses, (iii) distinguish between systemic and occasional quality issues, and (iv) reason about the mechanisms by which identified quality issues may have arisen in the event log. We contend that this knowledge can be used to guide the data extraction and pre-processing stages of a process mining case study to properly align the data with the case study research questions.

## 1. Introduction

Process mining is a maturing discipline and includes a set of tools and techniques to visualise process-related data. Process mining is being increasingly applied to healthcare (e.g., Rojas et al.’s [1] report on 74 process mining papers/case studies in healthcare, Yang and Su’s [2] report on 37 studies from 2004–2013 that apply process mining to various aspects of clinical pathways). However, some studies have shown that electronic medical records suffer data quality issues [3,4,5] which may impact on their usefulness in research. While noting the importance of data quality, existing process mining methodologies (i) do not provide details on how to assess the quality of event data (ii) do not consider how the identification of data quality issues can be exploited in the planning, data extraction and log building phases of any process mining analysis, (iii) do not highlight potential impacts of poor quality data on different types of process analyses.

The delivery of appropriate and timely pre-hospital care and transport of seriously injured road trauma patients is critical to patient survival and outcomes. Pre-hospital care and transport can be supplied by ground-based services, by aero-medical services, or by a combination of these two services. Both types of service are costly, resource intensive, asset limited and take significant coordination to deploy. In Queensland, road transport services are provided by the Queensland Ambulance Service (QAS) while clinical coordination of aero-medical services is managed by Retrieval Services Queensland (RSQ). RSQ utilises several providers for the supply of fixed wing and rotary aircraft. Comparing the various transport modes and escort levels, etc. may lead to a better understanding of factors contributing to patient outcomes. However, no research examining the retrieval processes (from roadside to definitive care) for road-trauma patients has been done in the Queensland context, and there is only limited research internationally.

The Queensland Trauma Plan in 2006 [6,7] prompted a review and restructure of trauma service delivery, and central coordination of trauma patient retrieval. Delivery of patients to the most appropriate service, in the most timely manner, was a key objective of the restructure. It has now been over 10 years since the trauma plan was released yet there hasn’t been a systematic review of the patterns of patient retrieval and the degree to which patient retrieval processes match desirable protocols. It is proposed to apply process mining techniques to:
discover the range of different care and transport processes undertaken for road trauma patients from roadside to definitive care for cohorts of incidents, patients and transports identified from the guidelines;conduct conformance (to guidelines) and comparative performance analyses for the discovered care and transport variants;identify key factors influencing deviance from standard care and delivery processes as given in the guidelines (e.g., patient demographics, patient injury types, mechanisms of injury), geospatial factors (of crash location and trauma facilities), responder characteristics (road vs aeromedical, paramedic/clinician attended crashes), etc.

This paper is an extension of our workshop paper [5] and differs from [5] in that we provide (i) a detailed description of the individual steps of our approach (including outputs and benefits derived from each step), (ii) a more detailed case study with additional process mining analyses, and (iii) an expanded related work section in which we consider a broader cross-section of process mining methodologies (particularly those that have either/both a healthcare or data quality focus). In this paper we present a process-centric, data quality-driven approach to preparing for a process mining analysis which can be applied to any existing process mining methodology. Our approach, adapted from elements of CRISP-DM [8], includes conceptual data modeling, quality assessment at both attribute and event level, and trial discovery and conformance to develop understanding of system processes and data properties to inform data extraction. We illustrate our approach in a case study describing the detailed preparation for a process mining analysis of ground and aero-medical pre-hospital transport processes involving the Queensland Ambulance Service (QAS) and Retrieval Services Queensland (RSQ). We utilise sample datasets from QAS and RSQ to show how data quality metrics, data models and exploratory process mining analyses can be used to (i) identify data quality issues, (ii) anticipate and explain certain observable features in process mining analyses, (iii) distinguish between systemic and occasional quality issues, and (iv) reason about the mechanisms by which identified quality issues may have arisen in the event log.

The major contributions of this paper include:
a contribution to the knowledge of how to establish a process mining study with particular emphasis on systematically identifying data-related issues prior to carrying out a process mining analysis;demonstration of how data quality issues manifest in process discovery and conformance analyses;a practical demonstration of the approach through a case study resulting in:
–conceptual data models (Object-Role Models (ORM)) of data held by (i) a ground based ambulance service provider (QAS) and (ii) a coordinator of aero-medical retrieval and transport service provider (RSQ);–an assessment of the quality (fitness for purpose) of the QAS and RSQ data for process mining analysis;–the derivation of context factors likely to impact on the study from literature regarding multiple aspects of pre-hospital retrieval and transport by both ground and aero-medical transport modes;a contribution to the knowledge-base in relation to ground emergency services (GEMS) and helicopter emergency services (HEMS) dispatch processes in an Australian context.

The remainder of this paper is organised as follows. In Section 2 we describe the outcomes of a literature review dealing with process mining in the healthcare domain, process mining methodologies and context factors that influence prehospital retrieval processes. In Section 3 we describe our approach to establishing a process mining case study. In Section 4 we illustrate our approach by applying it to the retrieval/transport process in Queensland. In Section 5 we discuss the data quality issues uncovered by our approach with implications for data collection for the full study. We conclude by highlighting the value of applying our approach in the early stages of a process mining analysis.

## 2. Related Literature

We considered literature relating to process mining in the healthcare domain (and in pre-hospital transport in particular). We look also at process mining methodologies and find that existing methodologies do not highlight the value of data quality assessment in the early stages of a process mining exercise. In considering articles dealing with pre-hospital retrieval, transport and pre-hospital care, we examined context factors affecting transport mode, destination/receiving hospital/trauma centre, and patient outcome. We found that little work has been done in applying process mining techniques to analyse pre-hospital processes.

### 2.1. Process Mining and Healthcare

Mans et al. [9] discuss event data recorded in Hospital Information Systems (HIS) and introduce the Healthcare Reference Model, a comprehensive data model designed to allow analysts to locate event data easily and to support data extraction. Rojas et al. [1] review 74 articles describing applications of process mining in the healthcare domain. Papers were characterised according to 11 points of relevance including process type, data type, frequently asked questions, analysis perspectives, tools, and methodologies. The authors conclude that future work should focus on the implementation of process-aware hospital information systems along with improved visualisation and visual analytics techniques and an increased focus on conformance checking in case studies. Yang and Su [2] report on 37 studies from 2004–2013 that apply process mining to various aspects of clinical pathways, classifying each study according to whether its major focus was process discovery, variants analysis and control, or evaluation and improvement. Andrews et al. [10] discuss the application of process mining techniques in the analysis of healthcare process-related data focussing on data extraction, pre-processing and data quality assessment. Consideration was given to challenges facing analysts in dealing with the semi-structured nature of healthcare processes when conducting discovery, conformance (comparative) performance analysis before providing some novel visualisation options. Bose et al. [11] describe a framework for classifying event log data quality issues according to the log entity affected (e.g., case, event, timestamp, etc.) and the type of quality issue—*missing, incorrect, imprecise, irrelevant* data. Fox et al. [12] implement Care Pathways Data Quality Framework (CP-DQF) which uses the framework described in [11] to support systematic management (identification, recording, mitigation, reporting) of data quality issues affecting process mining research using electronic healthcare records. Little work has been done in applying process mining techniques to analyse pre-hospital processes. Lamine et al. [13] apply process mining and discrete event simulation to assess the efficiency of emergency call centre operations in France and [14] apply process discovery, conformance checking and performance analysis in a case study involving ambulance services in Iran.

### 2.2. Process Mining Methodologies

In [15] the authors propose a methodology for process re-design (to overcome performance issues) that combines process mining and simulation. The methodology includes a data pre-processing step (conducted in collaboration with domain experts) aimed at bringing event data to a consistent level of aggregation. The authors note that even after event aggregation, the data quality may be uncertain and recommend the use of an algorithm tolerant to noise and exceptions to discover a process model from logged data (rather than identifying and addressing data quality issues).

Bozkaya et al.’s [16] Process Diagnostic Method (PDM) does not rely on prior or domain knowledge, and assumes that all information relevant to the analysis is contained in the event log (Figure 1). As a consequence, the authors explicitly state that the analysis will yield only facts about the process, with the responsibility for interpreting the results left to the organisation. PDM consists of 6 steps as:
**Step** **1****Log preparation** in which the event log is extracted from the organisation’s information systems.**Step** **2****Log inspection** from which a basic understanding of the process is developed.**Step** **3****Control flow analysis** which involves either checking that the event log conforms to an existing process description/model or automatically discovering a process model from the log.**Step** **4****Performance analysis** including discovering bottlenecks and calculating processing times.**Step** **5****Role analysis** which provides information on the division of work within the organisation (as it relates to the process being analysed).**Step** **6****Transfer of results** to process owners in such a way as they understand the outcomes, thus allowing the organisation to implement process changes.

Rebuge and Ferreira [17] modify PDM to include a Sequence Clustering step following Log Inspection that is specifically designed to identify process variants and infrequent behaviour frequently found in medical processes.

In the process Mining Manifesto [18] the authors outline the steps of the L* life-cycle model for conducting a process mining analysis (Figure 2). The L* model consists of 5 stages as:
**Step** **0****Justification and planning** to clearly outline the reasoning behind the study and to identify resources required for the study.**Step** **1****Extract** domain knowledge (from domain experts and historical data) to (i) develop an understanding of the domain and of the data available for analysis and, (ii) generate artifacts such as hand-made models, objectives and questions.**Step** **2****Create control-flow model and connect to event log** using automated discovery techniques.**Step** **3****Create integrated process model** by extending to other perspectives**Step** **4****Provide operational support** based on insights derived from earlier stages.

In [19] the author sets out to develop a process mining methodology that is business driven, is appropriate for all processes, and overcomes some perceived limitation of L*, i.e., that L* “describes the typical life-cycle for mining structured processes” [19] (p. 7) and that “L* presumes that a process mining analysis always has to start with a control-flow model” [19] (p. 7). The resulting Process Mining Project Methodology (PMPM) consists of 6 high-level functional requirements encompassing 18 required activities as (Figure 3):
**Step** **1****Scoping** Identify the process and gather basic knowledge; Determine the objectives of the project; Determine the required tools and techniques.**Step** **2****Data understanding** Locate the required data in the system’s logs; Explore the data in the system’s logs; Verify the data in the system’s logs**Step** **3****Event log creation** Select the dataset in terms of event context, timeframe and aspects; Extract the set of required data; Prepare the extracted dataset, by cleaning, constructing, merging and formatting the data**Step** **4****Process mining** Get familiar with the log by gathering statistics; Make sure that the process contained in the event log is structured enough to apply the required process mining techniques; Apply process mining techniques to answer business questions**Step** **5****Evaluation** Verify the modelled work; Validate the modelled work; Validate the modelled work; Decide on an elaboration of the process mining project**Step** **6****Deployment** Identify if and how the process can be improved by improvement actions; Present the project results to the organization

PMPM in Stage 2-Data understanding (Verify the data in the system logs) refers to assessing the quality of the data in the system logs against the quality attributes in [18] (Trustworthy, Completeness, Semantics, Safeness) but provides no guidance as to how such assessment should be made. Of importance is the guidance in Stage 3-Event log creation (Select the dataset in terms of …) in selecting events that are (i) relevant to the objectives of the study (ii) according to case completion, timeframe and data aspects.

The PM${}^{2}$ methodology [20] consists of 6 stages as:
**Step** **1****Planning** sets up the project and establishes research questions.**Step** **2****Extraction** identifies and extracts relevant data and, optionally, process models (if they exist). This stage is informed by the research questions identified earlier.**Step** **3****Data processing** has to do with creating event logs from the data specific to individual research questions.**Step** **4****Mining and analysis** applies process mining techniques to the event logs and aims to answer research questions and gain insights into process performance and compliance.**Step** **5****Evaluation** relates the analysis findings to improvement ideas that achieve the project’s goals.**Step** **6****Process improvement and support** implements the change notions derived in the previous stage and provides support by using process mining techniques to detect problematic running cases, predicting outcomes of running process instances or suggesting change actions for running cases.

In [20], the authors implement the PM${}^{2}$ methodology in a case study. They note several issues arising and lessons learned, one of which was the advisability of checking event data for errors. The authors note that failure to check event data quality early in the study led to incorrect results and necessitated repeating the data extraction and analysis.

It is worth noting that none of PDM, L*, PMPM or PM${}^{2}$ make explicit reference to, or include step(s) to identify (event) data quality issues or its effect on subsequent analysis.

In the healthcare domain, various process mining methodologies have been proposed [21,22,23,24]. In [22] the authors describe an interactive pattern recognition methodology and purpose-built tool that uses process mining techniques and event data captured from an RFID-based indoor location systems (ILS) with the aim of improving the surgical resource utilisation. As such, it is not a generally applicable process mining methodology. In [24] the authors describe the ClearPath method which extends the PM${}^{2}$ process mining methodology to include (healthcare) process simulation with the aim of addressing issues of poor quality and missing data through the use of CP-DQF (Care Path Data Quality Framework) designed for process mining of electronic health record (EHR) data [12]. The CP-DQF framework uses the dimensions in [11] for categorising data quality issues. In [12] the CP-DQF framework is implemented as a relational database and is used to maintain a registry of data quality issues, i.e., CP-DQF entries refer to data quality issue(s), include (SQL) code to identify and mark affected research data rows with the scale (severity) with which the issue affects the rows, and, where possible, includes mitigation code. At the time of writing, the CP-DQF registry was managed manually. That is, data quality issues affecting the EHR data were identified from a variety of sources, and, for each such issue, CP-DQF entries were created.

### 2.3. Context Factors Relevant to EMS Operations and Patient Outcomes

Here we identify factors, in the context of pre-hospital retrieval, that may be useful in partitioning cases into cohorts for separate analysis. In [25], the authors aimed to analyse the association between the use of physician-staffed helicopters versus ground services and survival among adults with serious traumatic injuries, and include the following independent variables: age; sex; initial vital signs (including SBP, RR and HR); mechanism of injury; type of trauma; transport mode; ISS (>15); and prehospital treatment, including intubation, airway protection maneuver and intravenous fluid. The study concluded that among patients with major trauma in Japan, transport by helicopter with a physician may be associated with improved survival to hospital discharge compared to ground emergency services. Leeuweberg and Hall [26] and Starnes et al. [27] in considering location of trauma incidents point out that distance (to trauma centre) is a factor in provision of emergency care with up to 65% of national road fatalities occurring in rural Australia [28] and that the time from trauma to first responder as being related to risk of death (19% increased risk of death per hour [29]). In terms of utility, Abe et al. [25] point out that aero-medical transports are affected by both weather and time of day (night time operations carry increased risk). Andrew et al. [30] in studying HEMS transport in Victoria, Australia found seasonality issues (transports more frequent on weekends and in summer).

## 3. Our Approach to Project Establishment and Assessing Data Quality

The impact of data quality on process mining analyses is well recognised [11,31]. However, existing process mining methodologies [16,17,18,20] do not specifically include consideration of the characteristics and quality of the input data in the early stages of a process mining study, i.e., prior to conducting the full analysis. Furthermore, there is little guidance on how to actually assess process-data quality. To address this issue, we adapt the CRISP-DM (Cross Industry Standard Process for Data Mining as outlined in [8], Figure 4) to systematically identify data quality issues and to link identified data quality issues to effects on process mining (particularly automated process discovery). The aim of the approach is to inform the final data extraction in such a way that identified quality issues can be addressed in the data that is extracted for the study proper. The steps in our approach include:
**Step** **1****Process Understanding** develop an understanding of the process to be investigated through interviews with stakeholders, from existing process models and standard operating procedures and manuals.
**Output**—High level ‘as-is’ process model**Benefits**—The model provides a point of contact to communicate, to the process owners, our understanding of the current process. The model also reveals various control-flow considerations such as event-ordering relationships.**Step** **2****Data Understanding** develop an understanding of the data available from all available sources, e.g., data dictionaries
**Output**—Conceptual data model/s**Benefits**—The conceptual data models provide an abstraction/idealisation of the actual data and are thus free from dealing with implementation related issues such as correlation issues across multiple information systems, data entry errors, noise, etc. Conceptual models also include cardinality relationships and constraints between data elements useful in event log construction.**Step** **3****Data Attribute Quality** quantify the quality of the sample data, at the attribute level, over several dimensions
**Output**—*n* x *m* matrix of *n* attributes and *m* quality dimensions where each cell stores a [0..1] value representing the quality of an attribute in the relevant dimension according to the metric used. For instance, the *completeness* dimension measures the fraction of records for which a given attribute has a value recorded.**Benefits**—Each quality measure can be used to (i) determine the suitability for an attribute’s inclusion/role in an event log, or (ii) anticipate the manifestation of certain structures or behaviours in process models derived form the data. For instance, an attribute that has a high *uniqueness* value would be unsuitable for inclusion in an event log as an activity label (discovered models would be highly complex). Mixed levels of *numeric precision* across attributes of datetime data type would lead us to anticipate event ordering issues in discovered process models (at least potentially unnecessary parallelism).**Step** **4****Event Log Preparation****Output**—Event log for use in pre-study process mining**Benefits**—The inclusion of data attributes and the assignment of same to event log attributes is informed by the quality values.**Step** **5****Event Quality** assess the quality of the event log derived in the previous step from the sample data, at the event level, by checking for the presence of Event Log Imperfection patterns [31]
**Output**—List of Event Log Patterns and whether or not the patterns exist in the data**Benefits**—Identification of the existence of one or more of the Event Log Imperfection patterns points immediately to the associated quality issue/s and impacts on process mining described, for each pattern in [31].**Step** **6****Pre-study Process Mining** conduct initial process mining analyses (e.g., discovery, conformance)
**Output**—At least, discovered process models**Benefits**—Allows checking for the presence of the structures/behaviours anticipated from the discovered quality issues. Conformance checking can reveal the extent of the issues in the log derived in Step 4.**Step** **7****Evaluation and Feedback** use the identified quality issues and quality metrics to:
inform our understanding of the process being investigated;revise, with process owners, questions to be investigated in the process mining study, data to be used in the study; andultimately, guide event log preparation for the study proper.

We see that this approach could be easily appended to existing process mining methodologies (during the planning phase but prior to the data extraction phase).

## 4. Illustration of Our Approach

In this section, we demonstrate our approach by applying it in a case study (pre-hospital retrieval and transport of road trauma patients in Queensland) to:
gain domain knowledge and an understanding of Queensland Ambulance Service’s notification-dispatch-retrieval-transport processes including articulation with Retrieval Service Queensland;gain an understanding of QAS and RSQ data through development of data models and examination of sample data extracts;conduct data quality assessment of each data set;prepare event logs relevant to the study aims;use the sample data to discover models of the individual QAS and RSQ retrieval/transport processes; andevaluate/conformance check the models.

### 4.1. Process Understanding—High-Level Patients Retrieval/Transport Process in Queensland

Figure 5 is a high-level retrieval/transport model derived from operating guideline documents provided by QAS and RSQ. All emergency calls (calls to 000) in Queensland are routed through a statewide network of 8 linked call centres operated by Queensland Ambulance Service. The QAS emergency centre operators gather as much information about the incident as they can from the caller reporting the incident. For road traffic or vehicle related incidents, this information includes (i) injury patterns, (ii) injury mechanisms, and (iii) vital signs. The combination of these three factors is used to determine the level of response required to deal with the incident. Usually, QAS will dispatch one or more ground-based ambulances to the scene of the incident, but, where the reported injury patterns, injury mechanisms and vital signs meet the criteria for Early Notification of Trauma (ENOT), the emergency operator will notify RSQ that aero-medical evacuation of injured person(s) may be required. ENOT criteria include ANY of
**Injuries** including (i) all penetrating injuries, (ii) significant blunt injuries to head, neck, chest, abdomen, pelvis, or axilla, (or injuries to multiple regions (iii) limb amputation, (iv) spinal chord injuries, (v) pelvis or lower limb compound fractures**Mechanisms of Injury** including (i) vehicle rollover, (ii) ejection from vehicle, (iii) fatality in same vehicle, (iv) impact >30 kph (motorcyclist) or >60 kph (other), (v) pedestrian impact, (vi) prolonged extrication**Vital Signs** (i) Resp. rate, (ii) Oxygen saturation, (iii) Systolic BP, (iv) Pulse rate, (v) Glascow Coma Score (with threshold values according to patient age

Once on-scene, QAS paramedics (i) will provide first-level support to injured patients, (ii) may contact a senior on-call paramedic or QAS Medical Coordinator (an experienced emergency doctor) for treatment advice, (iii) where the situation fits guidelines, may request aero-medical evacuation of injured person(s), or (iv) may transport patient directly by road. In cases of road transport, where clinical needs dictate, transporting paramedics may inform the receiving hospital of the patient’s condition and estimated arrival time.

Where aero-medical retrieval/transport is required, the QAS Communications Centre Supervisor (CCS) calls the RSQ Communication Centre. The call is picked up by a QAS Emergency Medical Dispatcher (EMD) stationed at the dedicated Rotary Wing Desk within the RSQ Communication Centre. The EMD has access to the statewide QAS Computer Aided Dispatch (CAD) which shows the Incident record. The EMD links the QAS CCS with the RSQ Medical Coordinator who discuss the incident and determine the optimal response. If the decision is made to dispatch an aircraft, the EMD tasks the aircraft while the RSQ Medical Coordinator contacts the retrieval team to fly on the respective aircraft and provides the patient’s clinical details. On arrival at the scene of the incident, or following contact with the patient, the retrieval team contacts the RSQ Medical Coordinator (via satellite phone or mobile phone). The RSQ Medical Coordinator provides specialist advice to, and oversight of, the retrieval team. They then determine the receiving hospital based on the patient’s clinical needs and informs the receiving, on-duty Emergency Department Specialist of the incoming patient, their estimated time of arrival and their clinical condition and requirements. On arrival at the Receiving Hospital, the retrieval team hands-over to the Emergency Department Specialist.

#### Scenarios

From the process description, high level BPMN model, data models, and discussions with domain experts, it is possible to derive some scenarios which may play out in response to any incident:
GEMS only, single/multiple patient, single/multiple response units dispatched, patient/s treated at scene, transport not required.GEMS only, single/multiple patient, single/multiple response units dispatched, patient/s treated at scene, transport provided by at least one response unit for at least one patient.HEMS primary response involving transport of a patient to closest hospital or, a regional trauma centre or, a major trauma centre.HEMS IFT (Inter-facility Transport) involving transport of a trauma patient initially delivered to closest hospital or a regional trauma centre who after some period, now requires transport to a major trauma centre.Fixed-wing primary response involving transport of a patient to closest hospital or, a regional trauma centre or, a major trauma centre.Multi-mode (GEMS + HEMS or Fixed-wing aero-medical) transport of trauma patient to closest hospital or, a regional trauma centre or, a major trauma centre.

In this preliminary study, only scenarios 1–3 are considered. The full study will consider all scenarios.

### 4.2. Data Understanding—Conceptual Data Models Relevant to Ground and Aero-medical Retrieval/Transport

From our understanding of emergency incident reporting-to-retrieval/transport (developed through interviews with domain experts, documentation describing QAS and RSQ data and informed by our literature review) we identified data relevant to the study that allows (i) end-to-end traceability (notification to delivery to definitive care) and which allows segmentation of the data into cohorts of retrieval/transport cases of interest to the process stakeholders. The Object-Role Models [32] in Figure 6 and Figure 7 depict the main data attributes necessary to allow end-to-end traceability and case segmentation for QAS and RSQ respectively. The main categories of data are as follows:
**Incident data** such as location of the incident, notification datetime the incident was reported to the emergency call centre and the priority of the incident.**Patient data** including patient name, age, gender, pre-existing conditions, allergies, current medications and indigenous status.**Transport data** which includes timestamped way-point data representing key case milestones, details of assessment of the scene, patient and injury by the paramedics, observations of the patient, management activities and procedures carried out by the ground-based paramedics or aircraft medical team, the destination hospital, and the patient outcome.

#### ORM Notation

For those readers not familiar with ORM notation we include a brief discussion of key features of the ORM notation used in this paper. An ORM model captures **relationships between entities**. An *entity type* is depicted as a round-cornered rectangle. An entity type has an *identifier label type* (e.g., Incident id). An entity type can have relationships with one or more entity types where a relationship is defined by the number of participating entities and the *role* played be each participating entity type. For instance, “suffers from” is a binary fact type (two participating entity types) between a Patient and a Pre-existing Condition and is modeled using a fact type with two roles connecting the entities involved. A bar above a fact type represents a *uniqueness constraint* that applies to that fact type. Uniqueness constraints may be one-to-one, one-to-many or many-to-many for binary fact types. We also indicate, through the *subset constraint* that a patient being transported by a vehicle dispatched in response to a particular incident must have been involved in the incident. A black dot attached to the connector between entity and role indicates that this role is *mandatory* for the entity. For instance every Incident must have a Location. An entity type can be associated with one or more *value types* which are depicted as a rectangle with dashed border (e.g., a Patient has a Name).

### 4.3. Data Attribute Quality Assessment

Intuitively, “high” quality data is more desirable than “poor” quality data as input for any data analysis. High quality data has been defined as “data that is fit for use by data consumers” [33] with data quality considered as “the degree to which the characteristics of data satisfy stated and implied needs when used under specified conditions” [34]. Operationalising this definition requires metric(s) to quantify the alignment between the characteristics of the data and the usage needs. Wand and Wang [35] state that data quality is a multi-dimensional concept where (i) each dimension represents some *measurable* quality property, and (ii) no single dimension can adequately assess overall data quality. Frequently mentioned data quality dimensions include accuracy/correctness, completeness, unambiguity/understandability and timeliness/currentness [34,35,36]. For this study, we use 3 (Completeness, Precision, Uniqueness) of the 20 quality dimensions frequently mentioned in [37] and their associated metrics. Using these metrics allows deriving insights into not only the state of the data in a particular column, but also into some possible impacts on process modeling. For instance, low values for the Precision metric [35] for datetime columns indicates coarse granularity (e.g., some values in the column may be at day level granularity). Coarse granularity of timestamps presents some issues from a process mining perspective as events may not be able to be properly sequenced (day level granularity events will always appear to occur before milli-second level granularity events for events that have the same date). The Completeness metric [38] measures the fraction of the rows of the data set for which a non-null value is recorded in the column. The Completeness metric then gives an indication of the suitability of the column for inclusion in an event log. For instance, if the column values are intended to be used to differentiate between cohorts of cases and the column is only 25% complete, it will not be possible to properly segment the set of cases. Lastly, the Uniqueness metric [39] provides a measure of the similarity of values in the column. For datetime columns that represent event log times, a high level of uniqueness is desirable. On the other hand, for columns that represent activity labels, a high level of uniqueness is undesirable. To conduct the assessment we (i) loaded the sample data into a relational database, then (ii) applied the column level quality checks, and (iii) checked for the presence of event log imperfection patterns as described in [31].

#### 4.3.1. QAS Sample Data

QAS provided a sample of 500 de-identified incidents attended by QAS between 01-July-2016 and 09-Jul-2016. The incident data was compiled from QAS’s Computer Aided Dispatch (CAD) system (used to allocate/dispatch vehicles to an incident) and its Clinical Information System (VACIS). The CAD system records the datetime of incident notification, (first) vehicle assignment, vehicle arrival on scene, departure from the scene, arrival at destination (hospital) and finally completion of the assignment. Not all waypoint times are recorded for vehicles not involved in a patient transport. The VACIS generates an Electronic Ambulance Report Form (eARF) which records, as well as clinical aspects, waypoint times for individual patients including vehicle en route, arrival at the scene, paramedics at the patient, patient loaded (for transport) and patient off-load (at hospital). Again, as not all attendances result in a transport, not all fields are populated.

The (tabular) data consisted of 15 columns, each of which represented an attribute of the attendance and patient transport. Attributes included an incident identifier, eARF number (proxy patient identifier) and patient and vehicle waypoint times. The data set contained 12 datetime type columns. From a process mining perspective, this gives, at most, 12 different activities that can be extracted from the data (with 2 waypoint times, one from the CAD system and one from the eARF that likely represent the same event—that is, the ‘At Scene’ event). The QAS domain expert indicated that, for any incident, multiple units may attend, and multiple patients may be involved. It is therefore possible to consider the data from at least three different “case” perspectives, i.e., an **incident** may be considered as a case, each **patient** may be considered as a case, or each **response unit** may be considered as a case. After consulting with the domain experts, it was determined that each patient should be the subject of the case. For the purposes of this part of the study, it was decided that eARF could be treated as surrogate patients, i.e., the eARF number would be the case identifier.

Table 1 provides descriptions, in terms of the QAS incident notification/dispatch/transport process, of the database column names provided in the sample data. **N.B.** for analysis purposes we use the data column names provided in the sample data.

**N.B.** The QAS domain expert further indicated that for any incident, more than one response unit may provide treatment to any patient. Each response unit generates a separate eARF record for each patient treated, hence a patient may have more than one eARF record for any incident, (which means the eARF value can not be used to uniquely identify a patient in the sample data). After considerable cleaning to remove duplicate data, each row of the sample data represents a patient treatment in an incident. Some of the time stamps are standardised across all records relating to a given incident to reflect the ‘First Assigned’ time (that is, all vehicles attending an incident will have the same value for the FIRST_ASSIGNED_CAD waypoint time). Others, such as On Scene/Depart Scene/Destination/Clear reflect the times for that specific unit. Not all timestamps are relevant to all attending units, hence some are empty e.g., D_LOADED_VACIS time is not recorded for units not transporting a patient. After de-duping the data and matching VACIS and CAD records, it was possible to match 723 eARF (VACIS) records with response unit (CAD) records.

Table 2 provides values for three column-level metrics useful in assessing the quality of the de-duplicated data.

Here we note that:
the **Completeness** metric shows that only 4 of the datetime columns are 100% complete which indicates that in any incident, not all patient and vehicle waypoints are completed. In particular, the 50% complete value for OFF_STRETCHER_VACIS indicates that only half the patients involved in incidents required transport to hospital. (We note that this may include missing data for transported patients, i.e., the paramedics did not manually record the timestamps in the record.)the **Precision** metric (for datetime) values gives an indication of mixed granularity among the various timestamps.the **Uniqueness** metric gives an indication of the degree of distinct values found in the column. The FIRST_ASSIGNED_CAD value shows low Uniqueness indicating many repeated values. This reflects the QAS policy of assigning to all vehicles involved in an incident, the timestamp of the first vehicle assigned to attend the incident.

The distinctly different values of the Precision metric between the _CAD timestamps and _VACIS timestamps suggests a difference in granularity between the sets of timestamps. Investigation revealed that all the _VACIS timestamps were recorded at minute-level granularity while the _CAD timestamps were recorded at second-level granularity. The immediate effect of the mixed granularity on event ordering can be seen when considering two events that must, in reality, occur in a particular order, but which appear to happen in a different order (according to their timestamps). For instance, D_RECEIVED_CAD represents the time when QAS Call Centre is notified of an incident and D_EN_ROUTE_VACIS represents the time a response unit is recorded as travelling to the incident scene. There are 52 (out of 723) cases where the D_EN_ROUTE_VACIS time is earlier than the D_RECEIVED_CAD time, and in 49 of these cases, the two timestamps are the same to minute-level granularity (as one example, for N_EARF = 76507098, D_RECEIVED_CAD = 2016-07-05 07:34:08 and D_EN_ROUTE_VACIS = 2016-07-05 07:34:00). This fits the description of the ‘Inadvertent Time Travel’ log imperfection pattern described in [31]. As such, temporal ordering of events in the log no longer matches reality. If left unaddressed, process mining analysis will be affected as (i) discovered process models will likely show these two events in parallel rather than, as expected, in sequence, and (ii) performance analysis will show incorrect activity/case durations.

#### 4.3.2. RSQ Sample Data

RSQ provided a sample of 500 de-identified aero-medical transports with case dates between 01-Mar-2017 and 28-Apr-2017. Inter-hospital Transfers formed the bulk of of the cases comprising (419 flights) with the remainder comprising 78 Primary Response missions (i.e., involving transport from scene of incident to hospital) and 3 Search and Rescue missions. The (tabular) data set was structured such that each row represented a separate mission and each of the 128 columns represented an attribute of the mission. The data included 62 mission records where the Mechanism of Injury value was ‘Vehicle accident’(comprising 35 Inter-hospital Transfers and 27 Primary Response missions). The data set contained only 12 datetime type columns. From a process mining perspective, this gives, at most, 12 different activities that can be extracted from the data.

Table 3 provides descriptions, in terms of the QAS incident notification/dispatch/transport process, of the database column names provided in the sample data. **NB** for analysis purposes we use the data column names provided in the sample data.

Table 4 provides values for some column-level metrics useful in assessing the quality of the data.

Here we note that:
the **Completeness** metric shows that all values are populated for the date time columns, while only 25% of the records in the log have a value for the MECHANISM_OF_INJURY column;the **Precision** metric (for datetime) values shows generally uniform granularity among the timestamps. Only the DATE_RETRIEVAL_REQUESTED column is day-level granularity, while all other other datetime columns are at minute-level granularity.the **Uniqueness** metric gives an indication of the degree of distinct values found in the column. The DATE_RETRIEVAL_REQUESTED value shows low Uniqueness indicating many repeated values. This is not surprising given the narrow range of case dates (many cases on any given day). The SOURCE_ID column shows perfect uniqueness (every value different from all others), while Uniqueness value of 27% for the MECHANISM_OF_INJURY column is reflective of the value being populated from a limited set of allowed values (e.g., a pull-down on a form).

The datetime columns represent milestone events in a mission and are expected to be sequential. However, we note that there are several violations (see Table 5) of such ordering apparent in the sample data.

### 4.4. Pre-Study Process Mining Analysis

In this section we complete the quality analysis by (i) generating event logs from the sample respective data sets, and (ii) using PromLite 1.2 (http://www.promtools.org/doku.php?id=promlite12) to perform basic process discovery (Inductive Visual Miner [40] plugin) and conformance analysis (Multi-perspective Process Explorer [41] plugin) to check that the event logs are suitable for process mining.

#### 4.4.1. QAS Process Discovery and Conformance

The de-duplicated QAS sample data was used to generate an event log where cases in the log represent an individual patient attendance/transfer. For the purpose of the preliminary analysis, each eARF in the sample data as taken to represent a patient. In creating the event log (i) the N_EARF column was mapped to the event log Case Identifier attribute, (ii) events were created from each datetime column in the dataset by (a) mapping the column name to the Activity Label attribute value and (b) assigning the row value of the column to the Timestamp attribute value (see Figure 8).

A process model was discovered using the Inductive Visual Miner plugin in ProMLite 1.2 (shown filtered to 80% paths in Figure 9).

Conformance checking with the Multi-perspective Process Explorer plugin in ProMLite showed the model had 94.3% fitness (490 wrong and missing events out of 5,595 events in total). The activities highlighted in yellow in the conformance model shown in Figure 10 show where wrong events occur or missing events were expected. The discovered model highlighted some variations from the expected process behaviour as described by the QAS domain expert (illustrated in Figure 5) and also highlighted some of the data quality issues discussed in Section 4.3.1.

For instance, the expected behaviour is sequential execution of milestone tasks while the discovered model shows parallelism, (e.g., D_EN_ROUTE_VACIS and D_AT_SCENE_VACIS occur in a parallel block). Investigation showed that while there were no cases where D_AT_SCENE_VACIS preceded D_EN_ROUTE_VACIS, there were 14 cases where the timestamp values for these two activities were the same. As observed earlier, the data quality analysis precision metric for the _VACIS times indicated only minute-level granularity. This may represent a “field dispatch” (i.e., non-tasked ambulance encounters an accident and notifies EMD it is on-scene). As such, the milestone events actually occurred in the expected sequence, but very close together (i.e., within the same minute) such that the recorded values were identical. In a similar vein, investigating the parallelism exhibited around the D_ON_SCENE_CAD and D_AT_PAT_VACIS activities showed that for the 581 cases where both activities occurred, in 198 cases the D_AT_PAT_VACIS activity occurred before the D_ON_SCENE_CAD activity. However, 174 of these cases had timestamps within 1 minute of each other. Taking into account the minute-level granularity of the _VACIS times, it is again possible that these milestone events, in reality, occurred in the expected sequence, but very close together (i.e., within the same minute) but that the mixed granularity of the _VACIS and _CAD times results in incorrect event ordering. (We note that there were in fact 24 cases where there was ‘real’ deviation from the expected event ordering.)

Lastly, we note that the discovered model reflects the nature of the various types of attendance. For instance, (i) the 359 cases which skip the D_LOADED_VACIS and D_DEPART_SCENE_CAD steps reflect that not all attendances required the transport of a patient to hospital, and (ii) the 53 cases where a D_AT_DEST_CAD occurs without a corresponding D_OFF_STRETCHER_VACIS event may reflect a non-transporting unit (e.g., Critical Care Paramedic backup) has proceeded to the hospital to accompany the transporting unit.

#### 4.4.2. RSQ Process Discovery and Conformance

In a similar manner as for the QAS sample, the de-duplicated RSQ sample data was used to generate an event log. Again, the individual patient transfer was taken as the case, i.e., each row in the sample data was taken to represent a case. In creating the event log (i) the SOURCE_ID column was mapped to the event log Case Identifier attribute, and (ii) events were created from each datetime column in the dataset by (a) mapping the column name to the Activity Label attribute value and (b) assigning the row value of the column to the Timestamp attribute value. A process model (shown filtered to 90% paths in Figure 11) was discovered using the Inductive Visual Miner plugin in ProMLite 1.2. Conformance checking with the Multi-perspective Process Explorer plugin in ProMLite showed the model had 98.7% fitness (156 wrong and missing events out of 5922 events in total). The discovered model highlighted some variations from the expected process behaviour as described by the RSQ domain expert (illustrated in Figure 5) and also highlighted some of the data quality issues discussed in Section 4.3.2. The model shows parallelism for activities following AT_SCENE_PATIENT where the expected behaviour is sequential. The data quality assessment (see Table 5) and the model identified the activities and the extent of the deviation from expected behaviour. The activities highlighted in yellow in the conformance model shown in Figure 12 show where wrong events occur or missing events were expected. The conformance analysis revealed other event ordering issues including 10 cases where the first activity was not DATE_RETRIEVAL_REQUESTED.

## 5. Discussion and Lessons Learned

In creating an event log from non-PAIS source data where event ordering is on the basis of timestamps, *datetime* type fields are important as they become the markers for identifying events in the source data. Event construction then becomes a matching exercise; for each datetime column, find the source data columns/values that should be assigned to the various event attributes (e.g., case identifier, activity label, etc). In this case study, each source data record represents multiple events (that is one row in the source contains fields such as DATE_RETRIEVAL_REQUESTED and TEAM_ACTIVATED, each of which marks a milestone event in the case). When constructing event records from such data, each datetime field generates a separate event where the **name** of the datetime column is mapped to the value of the event activity label and the **value** of the datetime column is mapped to the value of the timestamp attribute.

The conceptual data modelling prior to process mining informs the data extraction phase of the case study. For instance, the respective QAS and RSQ data models reveal multiple 1:*n* relationships among entities involved in the prehospital retrieval/transport process. For instance, an incident may involve many patients, an incident may involve many response units, a response unit may provide prehospital treatment to more than one patient, and for any patient, injury to final discharge (or death) may involve multiple transport legs. Thus, there are many possible *case* perspectives that are relevant (i.e., an incident may be considered a case, an individual patient experience may be considered a case, an individual response unit’s dispatch/attendance/transport may be considered a case, etc). An important consideration in extracting the final dataset will be ensuring that, as well as the stakeholder’s view that the patient experience is the case perspective, it will be possible to investigate other case perspectives. We also note that the data model revealed that other activities were carried out during prehospital retrieval/transport but, as this data was not included in the sample data, it was not possible to determine when, or in which order, such activities took place.

The quality assessment of the (sample) data, conducted **prior** to the discovery and conformance analyses, adds value to the overall process mining exercise in at least **four** ways.
**Identifying event-data quality issues allows for the anticipation of certain observable features in subsequent process mining analysis.** For instance, the mixed granularity in timestamps apparent in the quality assessment (different values for the Precision metric) led the analysts to anticipate incorrect event ordering being an issue (subsequently confirmed by the parallelism apparent in the discovered models). Furthermore, the coarse granularity of the (RSQ) DATE_RETRIEVAL_REQUESTED values (all are at day-level granularity) precludes the possibility of properly assessing performance aspects of some phases of aero-medical retrieval (for instance, the time taken to activate a medical team following a retrieval request cannot be assessed as the retrieval request includes only a date with no time information). For the ground-based retrieval/transport data, the quality analysis showed duplication in the FIRST_ASSIGNED_CAD values. After discussion with QAS it emerged that it is QAS practice to include, for all response units dispatched to attend an incident, the same value for FIRST_ASSIGNED_CAD. From a QAS perspective, this allows assessment of time between incident notification and first response to the incident. As such, this is not a data quality issue (as it is done purposefully by the process owner) and can be taken into account by making this a case attribute. Identifying this issue through quality assessment headed-off issues that may have arisen in the process mining analysis had the FIRST_ASSIGNED_CAD milestone been included as an activity for all eARFs and response units involved in the incident.**Quantifying quality issues means that it is possible to separate systemic from occasional quality ‘breaches’.** For instance, the fact that all (QAS) _VACIS timestamps were at a low level of precision (i.e., minute-level granularity) points to a systemic cause.**Identifying quality issues allows for reasoning about the mechanisms that may have caused the quality issue to be present in the event data.** For instance, it is unlikely that **all** (QAS) _VACIS events happened exactly on the minute, but, it is likely that either the system recording the event had only minute-level precision or that in extracting the data for analysis, seconds and milli-seconds were ‘masked’. The fact that some (RSQ) cases have ARRIVE_AT_RECEIVING_HOSPITAL and DEPART_RECEIVING_HOSPITAL occurring at the same times may indicate a combination of human and system issues, i.e a human omission to record the ARRIVE time when the aircraft arrives (possibly due to patient care needs), and a system requirement that an ARRIVE time needs to be entered before a DEPART time can be entered.**An understanding of 2 and 3 above facilitates informed engagement with process stakeholders and decisions about data quality remediation actions.** For instance, if the _VACIS granularity issues were as a result of incorrect data extraction, this quality issue can be resolved by simply extracting the data at the appropriate granularity.

## 6. Conclusions

In this paper we have described an approach that supplements existing process mining methodologies by including steps explicitly designed to identify data quality issues in process-related data prior to constructing event logs and conducting the study proper. The approach focuses on identifying the quality issues and anticipating how the quality issues will impact a process mining analysis should the discovered issues not be addressed. Advantages of the approach include (i) early identification of quality issues, (ii) early identification of errors in discovered process models and performance analyses, (iii) improved understanding of the data being used to support the process mining study, and (iv) opportunity to revise data extraction and log generation (in the light of quality assessments) to minimise the risk of erroneous analysis (and consequent rework) due to poor data.

Currently, the approach is limited to the identification of data quality issues. Clearly, one avenue for future work will be to expand the approach to include a log repair step. We note that effective log repair would require the development of metric(s) to assess the impact of the repairs, i.e., to determine whether the repair actions truly result in improvements to the log quality.

## Figures and Tables

**Figure 1 ijerph-16-01138-f001:**
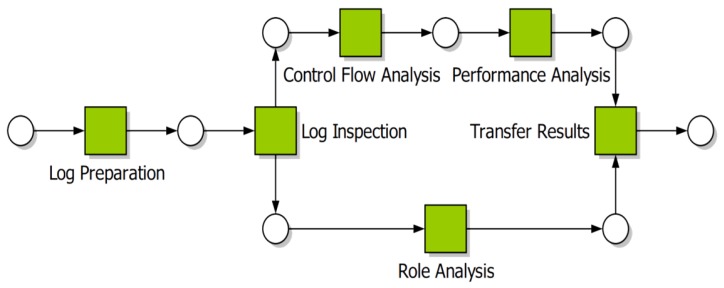
The PDM methodology [16].

**Figure 2 ijerph-16-01138-f002:**
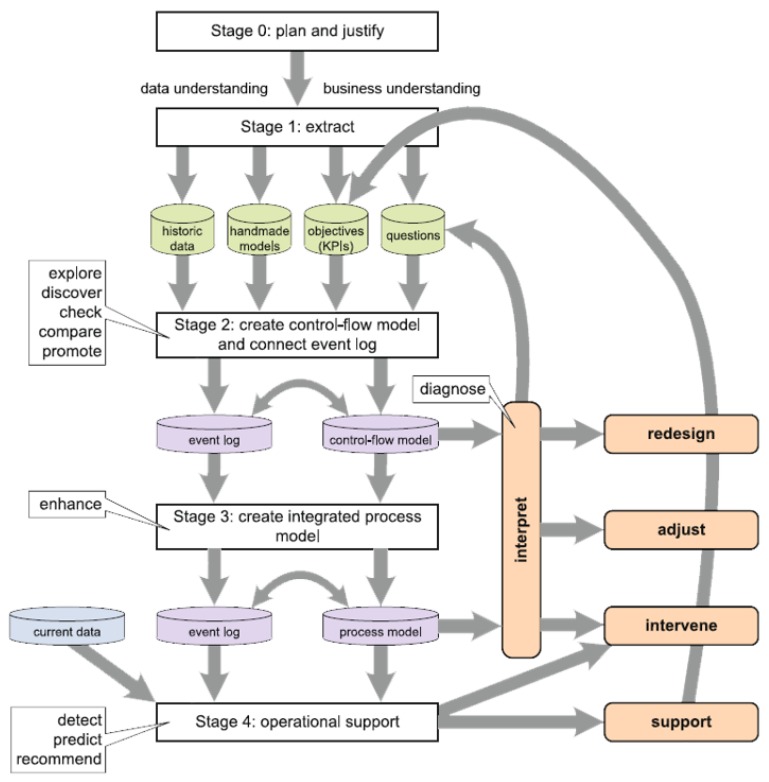
The L* methodology [18].

**Figure 3 ijerph-16-01138-f003:**
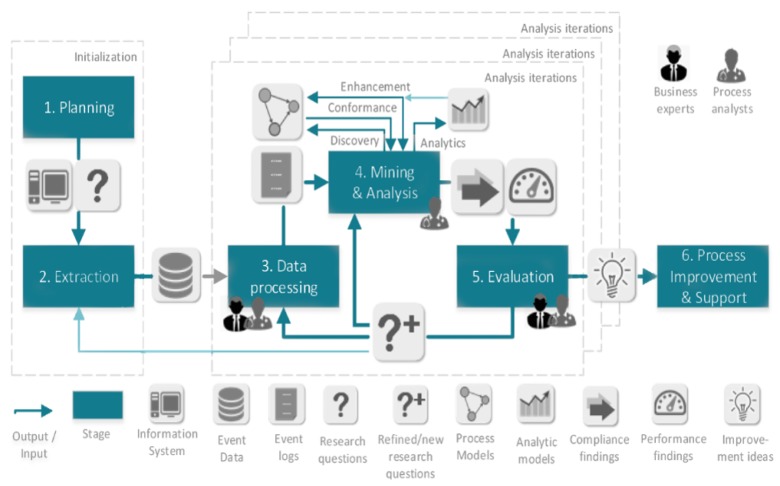
The PM${}^{2}$ methodology [20].

**Figure 4 ijerph-16-01138-f004:**
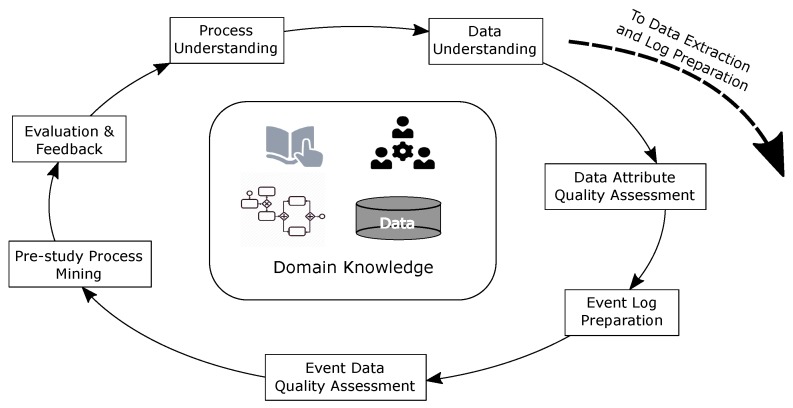
Our approach adapted from the CRISP-DM methodology [8].

**Figure 5 ijerph-16-01138-f005:**
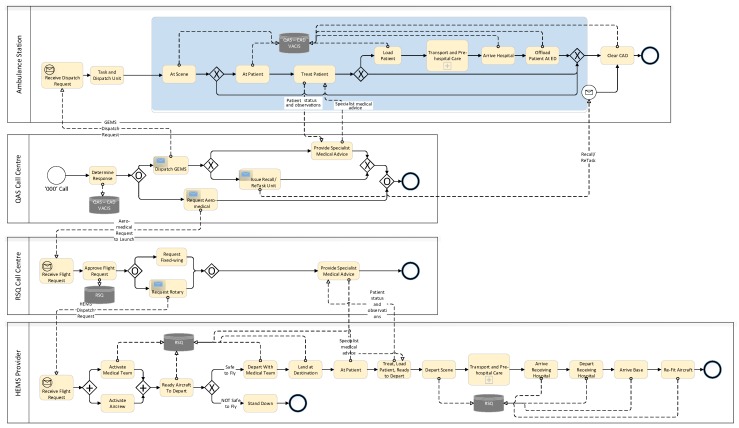
BPMN model of emergency incident management—ground and aero-medical call centre, asset deployment and patient transport.

**Figure 6 ijerph-16-01138-f006:**
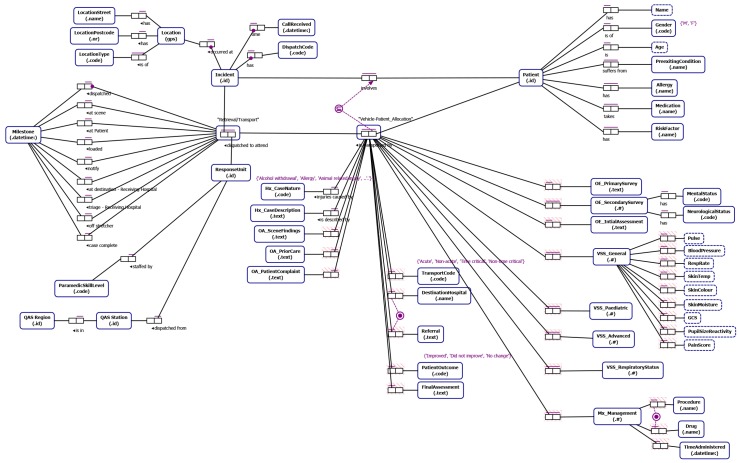
ORM model of QAS data.

**Figure 7 ijerph-16-01138-f007:**
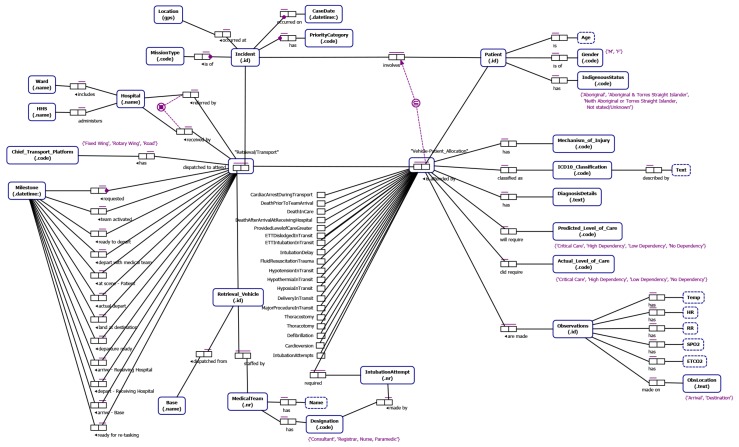
ORM model of RSQ data.

**Figure 8 ijerph-16-01138-f008:**
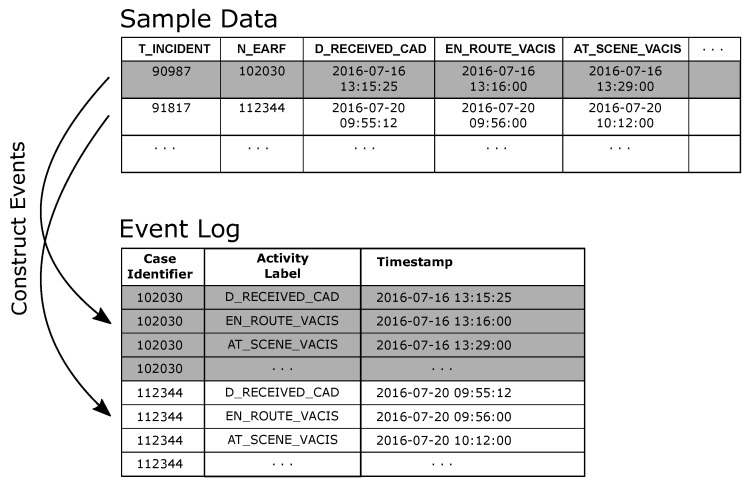
Event Log creation from sample data.

**Figure 9 ijerph-16-01138-f009:**
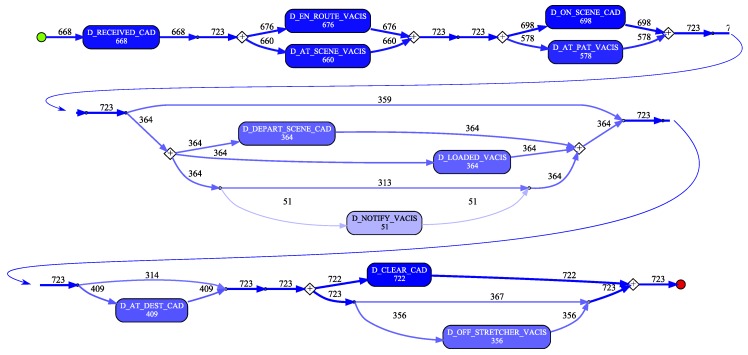
QAS process model derived from sample data.

**Figure 10 ijerph-16-01138-f010:**
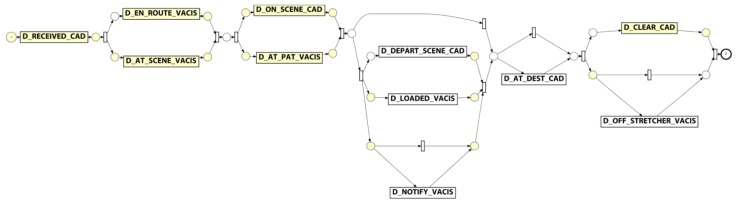
QAS conformance model derived from sample data.

**Figure 11 ijerph-16-01138-f011:**
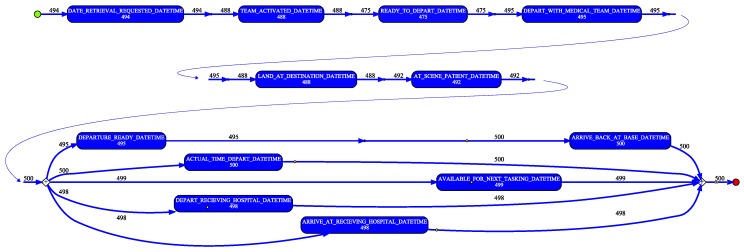
RSQ process model derived from sample data.

**Figure 12 ijerph-16-01138-f012:**
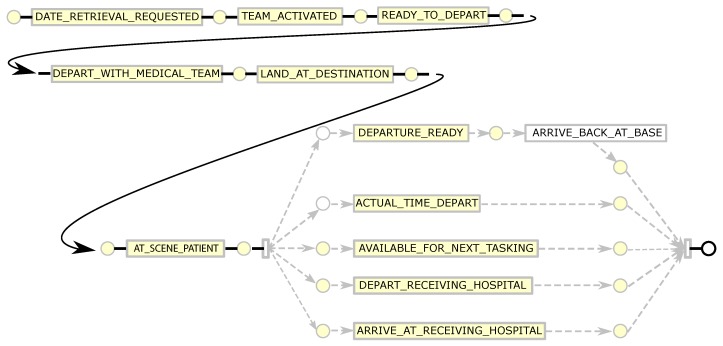
RSQ conformance model derived from sample data.

**Table 1 ijerph-16-01138-t001:** QAS—Data attributes and descriptions.

Column Name	Description
T_INCIDENT	Incident identifier
D_RECEIVED_CAD	Date/time the QAS emergency call centre is notified of an incident and a request for an ambulance.
FIRST_ASSIGNED_CAD	Date/time when the **first** ambulance unit is dispatched to attend the incident.
CAD (Vehicle) Waypoints	
ON_SCENE_CAD	Date/time when a unit arrives at the incident scene.
DEPART_SCENE_CAD	Date/time when a unit departs the incident scene.
AT_DEST_CAD	Date/time when a unit arrives at destination. This is usually a hospital emergency department.
CLEAR_CAD	Date/time when a unit indicates its involvement in the incident is finished (and is available for re-tasking).
eARF (Patient) Waypoints	
EN_ROUTE_VACIS	Date/time recorded by a unit indicating it has commenced travelling to the incident scene.
AT_SCENE_VACIS	Date/time recorded by a unit when it arrives at the incident scene.
AT_PAT_VACIS	Date/time recorded by a unit when paramedics arrive at a patient. May be different from arriving at the incident scene as the patient location may be inaccessible by vehicle necessitating the paramedics walk to the patient.
LOADED_VACIS	Date/time recorded by a unit when a patient is loaded in the unit ready for transport.
NOTIFY_VACIS	Date/time recorded by a unit on leaving incident scene.
OFF_STRETCHER_VACIS	Date/time recorded by a unit when a patient is unloaded from the unit (at destination).

**Table 2 ijerph-16-01138-t002:** QAS—Column-level data quality summary for a sample of columns.

Column Name	Data Type	Completeness	Precision	Uniqueness
T_INCIDENT	int	100%		100%
D_RECEIVED_CAD	datetime	100%	83%	100%
FIRST_ASSIGNED_CAD	datetime	100%	83%	58%
CAD (Vehicle) Waypoints				
ON_SCENE_CAD	datetime	97%	83%	83%
DEPART_SCENE_CAD	datetime	57%	83%	82%
AT_DEST_CAD	datetime	57%	83%	82%
CLEAR_CAD	datetime	100%	83%	83%
eARF (Patient) Waypoints				
EN_ROUTE_VACIS	datetime	94%	66%	84%
AT_SCENE_VACIS	datetime	95%	66%	86%
AT_PAT_VACIS	datetime	90%	66%	90%
LOADED_VACIS	datetime	52%	66%	89%
NOTIFY_VACIS	datetime	7%	66%	53%
OFF_STRETCHER_VACIS	datetime	50%	66%	93%

**Table 3 ijerph-16-01138-t003:** RSQ—Data attributes and descriptions.

Column Name	Description
SOURCE_ID	Incident identifier.
DATE_RETRIEVAL_REQUESTED	Date the RSQ is notified of an incident and of a request for aero-medical transport.
TEAM_ACTIVATED	Date/time when the medical team crewing is alerted to fly.
READY_TO_DEPART	Date/time when the unit is ready to depart base.
DEPART_WITH_MEDICAL_TEAM	Date/time when a unit actually departs base.
LAND_AT_DESTINATION	Date/time when a unit arrives at the incident scene.
AT_SCENE_PATIENT	Date/time when the medical team arrives at a patient. May be different from arriving at the incident scene as the patient location may be inaccessible by vehicle necessitating the paramedics walk to the patient.
DEPARTURE_READY	Date/time when a unit is ready to depart from the incident scene.
ACTUAL_TIME_DEPART	Date/time when a unit actually departs the incident scene.
ARRIVE_AT_RECIEVING_HOSPITAL	Date/time when a unit arrives at the receiving hospital.
DEPART_RECIEVING_HOSPITAL	Date/time when a unit departs from the receiving hospital (after handing over patient to hospital medical team.
ARRIVE_BACK_AT_BASE	Date/time when a unit returns to base.
AVAILABLE_FOR_NEXT_TASKING	Date/time when a unit is refitted and ready for re-tasking.
MECHANISM_OF_INJURY	mode of injury necessitating aero-medical rather then ground-based ambulance attendance.

**Table 4 ijerph-16-01138-t004:** RSQ—Column-level data quality summary for a sample of columns.

Column Name	Data Type	Completeness	Precision	Uniqueness
SOURCE_ID	int	100%		100%
DATE_RETRIEVAL_REQUESTED	datetime	100%	33%	7%
TEAM_ACTIVATED	datetime	100%	65%	95%
READY_TO_DEPART	datetime	100%	65%	96%
DEPART_WITH_MEDICAL_TEAM	datetime	100%	65%	95%
LAND_AT_DESTINATION	datetime	100%	65%	96%
AT_SCENE_PATIENT	datetime	100%	65%	97%
DEPARTURE_READY	datetime	100%	65%	96%
ACTUAL_TIME_DEPART	datetime	100%	65%	96%
ARRIVE_AT_RECIEVING_HOSPITAL	datetime	100%	65%	96%
DEPART_RECIEVING_HOSPITAL	datetime	100%	65%	96%
ARRIVE_BACK_AT_BASE	datetime	100%	66%	93%
AVAILABLE_FOR_NEXT_TASKING	datetime	100%	63%	92%
MECHANISM_OF_INJURY	string	25%	100%	27%

**Table 5 ijerph-16-01138-t005:** RSQ—milestone activity ordering violations.

Milestone Activities	*a* before *b*	a=b	*a* after *b*
a= DEPARTURE_READY b= ACTUAL_TIME_DEPART	434	66	0
a= ARRIVE_AT_RECEIVING_HOSPITAL b= DEPART_RECEIVING_HOSPITAL	470	29	1
a= ARRIVE_BACK_AT_BASE b= AVAILABLE_FOR_NEXT_TASKING	315	105	80

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
