# Peer review of "Leveraging Data Quality to Better Prepare for Process Mining: An Approach Illustrated Through Analysing Road Trauma Pre-Hospital Retrieval and Transport Processes in Queensland"

_ijerph, 2019, doi:10.3390/ijerph16071138_

Round 1

Reviewer 1 Report

The study presents a data quality-driven approach to prepare for a process mining analysis, it provides information about the quality issues normally present in data when applying process mining and building event logs. They include metrics and a very good detailed case study for ground and aerial transportation in Queensland. Data for the case study was extracted from two different sources, for each one including their respective data model.

The introduction is complete and contextualizes in a very nice way the study. Also includes the contributions and the description that the paper is an extended version. More clarity on what content in the paper is new compared to the previously published one.

The related literature is where I find that more information is needed. When the authors describe Process mining methodologies in point 2.2 they only focus on Bozkaya’s, Rebuge and Ferreira’s, The L* cycle model, and the PM2 methodology. The authors only cover these four, when in the literature, you can find at least five or more, related at least to the application of process mining in healthcare. Some of these, even address some of the quality data tasks and issues presented in this paper. I think it is important to at least mention them and discuss if these other ones propose something related to data quality management for process mining.

Some of the other methodologies are:

Măruşter, L., & van Beest, N. R. (2009). Redesigning business processes: a methodology based on simulation and process mining techniques. Knowledge and Information Systems, 21(3), 267.

Fernandez-Llatas, C.; Lizondo, A.; Monton, E.; Benedi, J.-M.; Traver, V. Process Mining Methodology for Health Process Tracking Using Real-Time Indoor Location Systems. Sensors 2015, 15, 29821-29840.

Johnson, O. A., Dhafari, T. B., Kurniati, A., Fox, F., & Rojas, E. (2018, September). The ClearPath Method for Care Pathway Process Mining and Simulation. In International Conference on Business Process Management (pp. 239-250). Springer, Cham.

Van Der Heijden, T. H. C. (2012). Process mining project methodology: Developing a general approach to apply process mining in practice. Master of Science in Operations Management and Logistics. Netherlands: TUE. School of Industrial Engineering.

Cho M., Song M., Yoo S. (2014) A Systematic Methodology for Outpatient Process Analysis Based on Process Mining. In: Ouyang C., Jung JY. (eds) Asia Pacific Business Process Management. AP-BPM 2014. Lecture Notes in Business Information Processing, vol 181. Springer, Cham

Rojas, E.; Sepúlveda, M.; Munoz-Gama, J.; Capurro, D.; Traver, V.; Fernandez-Llatas, C. Question-Driven Methodology for Analyzing Emergency Room Processes Using Process Mining. Appl. Sci. 2017, 7, 302.

There is even a Data quality framework for process mining in healthcare published last year. This study might not have been included in the original paper, but it should be addressed in this version.

Fox, F., Aggarwal, V. R., Whelton, H., & Johnson, O. (2018, June). A data quality framework for process mining of electronic health record data. In 2018 IEEE International Conference on Healthcare Informatics (ICHI) (pp. 12-21). IEEE.

On Section 3, the authors establish their approach to data quality adapting the CRISP-DM to process mining, including specific outputs and benefits for each of the steps in the process. This section is very clear and provides information in a simple and efficient way.

Section 4 provides the case study of the pre-hospital retrieval and transport of road trauma patients in Queensland, which is the most complete section, including from objectives, business context, BPMN models of the expected processes, multiple healthcare scenarios, data models, data quality analysis and metrics, to process mining discovery and conformance techniques. This Section is complete and requires no significant changes.

The remaining sections provide some analysis and discussion regarding the case study and the data quality analysis, which are clear.

In general, the study is well written and easy to follow and understand, it can provide a good guide for other researches that want to execute case studies using data extracted from non-Process Oriented systems. The only issue I have is the study of additional methodologies to see if some of those provide data quality analysis, and how they differentiate from the analysis done in this paper. Also, it is necessary to address how quality frameworks for process mining in healthcare compare to this.

Author Response

We thank Reviewer 1 for his/her comments and suggestions for revisions. The following details the actions we took to points raised by the reviewer as needing attention.

Introduction: The reviewer requested that more clarity on what content is new compared to the previously published version.

Response: We have added a description (bottom of page 2) that makes clear the differences between this paper and the previous workshop.

Related Literature: The reviewer provided a list of process mining methodologies and suggested it was important to mention these.

Response: Each method has been considered and described (or at least mentioned) in the text. Distinctions were made between general purpose process mining methodologies and those specifically aimed at healthcare. Where relevant, the methodologies approach to data quality was highlighted.

Related Literature: The reviewer mentioned a data quality framework for healthcare and process mining which should be addressed in the paper.

Response: The framework was described in the related literature section.

Reviewer 2 Report

The authors presents a strudy about how data Quality issues affect to Process Mining technologies.

The authors present related work about process Mining methodologies, and analize 3 quality issues in a healthcare real log.

The process Mining methodologies are presented, but there is not a comparison among them, for example a table making the comparison among them will make more readable the paper. Also a comparison step by step with the proposed methodology will be more clarifier.

In my humble opinion the explanation of the methodology is too much schematic and affects to readability of the paper. In my opinion, there is an abuse in the use of 'bullets' in the paper, but this is only a minor suggestion. 

The Abstract should be self-content. The references to bibliography should be avoided.

In my opinion the example is difficult to follow, The use of the log names (D_RECEIVED_CAD, T_INCIDENT...) affects negatively to the readability of the paper I suggest to change the names to more understandable ones

Author Response

We thank Reviewer 2 for his/her comments and suggestions for revisions. The following details the actions we took to points raised by the reviewer as needing attention.

Issue: The process Mining methodologies are presented, but there is not a comparison among them, for example a table making the comparison among them will make more readable the paper. Also a comparison step by step with the proposed methodology will be more clarifier.

Response: The intent of our paper is not to compare existing process mining methodologies. the Related Work section of the paper does indeed discuss many existing process mining methodologies, some in detail. The methodologies discussed are end-to-end analysis approaches, whereas the approach presented in our paper deals only with the data pre-processing stage of an analysis and focusses on specific ways in which data quality issues likely to affect a process mining analysis can be detected using a metrics-based approach. We do point out that referenced methodologies such as PDM, L*, PM2, PMPM refer obliquely to the importance of data quality but do not specifically state how to identify or quantify the scale of quality issues. Thus we address a clear gap in other methodologies and we say that our approach can be incorporated into any of the other methodologies. Therefore comparing each of the referenced methodologies with each other (or our approach) would seem to be outside the scope of this paper. 

Issue: The Abstract should be self-content. The references to bibliography should be avoided.

Response: The Abstract has been reworded to remove reference to the bibliography.

Issue: In my opinion the example is difficult to follow, The use of the log names (D_RECEIVED_CAD, T_INCIDENT...) affects negatively to the readability of the paper I suggest to change the names to more understandable ones

Response: The names used in the case study were those provided by the industry partners. To address the issue of making the case study more readable, we have included tables that provide a description of the meaning of each database column in the context of the incident notification/dispatch/transport process.

This manuscript is a resubmission of an earlier submission. The following is a list of the peer review reports and author responses from that submission.